# Resorbable Beads Provide Extended Release of Antifungal Medication: In Vitro and In Vivo Analyses

**DOI:** 10.3390/pharmaceutics11110550

**Published:** 2019-10-24

**Authors:** Yung-Heng Hsu, Huang-Yu Chen, Jin-Chung Chen, Yi-Hsun Yu, Ying-Chao Chou, Steve Wen-Neng Ueng, Shih-Jung Liu

**Affiliations:** 1Department of Orthopedic Surgery, Chang Gung Memorial Hospital-Linkou, Tao-Yuan 33305, Taiwan; laurencehsu.hsu@gmail.com (Y.-H.H.); alanyu1007@gmail.com (Y.-H.Y.); enjoycu@ms22.hinet.net (Y.-C.C.); 2Department of Mechanical Engineering, Chang Gung University, Tao-Yuan 33302, Taiwan; kyoiorikyo@gmail.com; 3Graduate Institute of Biomedical Science, Chang Gung University, Tao-Yuan 33302, Taiwan; Jinchen@mail.cgu.edu.tw

**Keywords:** fluconazole, orthopedic infection, Poly(d,l-lactide-*co*-glycolide) beads, sustained release

## Abstract

Fungal osteomyelitis has been difficult to treat, with first-line treatments consisting of implant excision, radical debridement, and local release of high-dose antifungal agents. Locally impregnated antifungal beads are another popular treatment option. This study aimed to develop biodegradable antifungal-agent-loaded Poly(d,l-lactide-*co*-glycolide) (PLGA) beads and evaluate the in vitro/in vivo release patterns of amphotericin B and fluconazole from the beads. Beads of different sizes were formed using a compression-molding method, and their morphology was evaluated via scanning electron microscopy. Intrabead incorporation of antifungal agents was evaluated via Fourier-transform infrared spectroscopy, and in vitro fluconazole liberation curves of PLGA beads were inspected via high-performance liquid chromatography. When we implanted the drug-incorporated beads into the bone cavity of rabbits, we found that a high level of fluconazole (beyond the minimum therapeutic concentration [MTC]) was released for more than 49 d in vivo. Our results indicate that compression-molded PLGA/fluconazole beads have potential applications in treating bone infections.

## 1. Introduction

Fungal osteomyelitis is an uncommon disease presenting significant challenges to orthopedic surgeons. The severity of a fungal infection is associated with the immune status of individuals and the fungal species. A large number of fungal infections have been reported in both immunocompromised and immunocompetent individuals [1]. Fungal infections typically result from three routes: direct inoculation, hematogenous spreading, or contiguous spreading [2,3]. In certain conditions, including long-term antibiotic use, infections of *Candida*, *Aspergillus*, and other common fungi are characterized by the formation of a biofilm that resists antifungal treatment, thus further strengthening the infection [4]. Certain fungal infections can exclusively be treated with antifungal agents [5]. However, most patients are difficult to treat via radical debridement. General principles for treatment of both bacterial osteomyelitis and fungal infections include excision of all nonviable tissue, the use of orthopedic hardware, radical debridement, and local drug delivery with an effective concentration of antifungal/antibiotic agents [6,7].

The most common fungi causing bone and joint infections are *Candida* spp., and *Candida albicans* in particular [8]. Fluconazole is a first-generation triazole antifungal agent commonly used to treat *Candida albicans* infections [9]. Furthermore, amphotericin B sodium deoxycholate effectively inhibits biofilm formation in multiple *Candida* infections [10]. Treatment of *Candida* osteomyelitis involves surgical debridement and long-term administration of antifungal agents. The guidelines of the Infectious Diseases Society of America (IDSA) for the treatment of *Candida* osteomyelitis suggest the administration of fluconazole for 6–12 months [11]. Currently, antifungal-agent-containing beads or polymethyl methacrylate (PMMA) spacers are popular treatment methods for fungal osteomyelitis or periprosthetic joint fungal infections [4]. The beads or spacers delivered to bone tissue display sustained long-term release of antifungal agents at an effective concentration. However, the potential risk of irritation in host tissues combined with the non-biodegradable nature of PMMA may limit application in osteomyelitis treatment, thus warranting surgical excision. Kweon et al. reported that adding 10 g poragen to antifungal-loaded bone cement (ALBC) containing 200 mg amphotericin B decreases the compressive strength of PMMA beads and thus limits their use for implant fixation [12]. Sealy et al. showed poor release dynamics of fluconazole in ALBC [13]. Furthermore, Goss et al. reported that amphotericin B could not be eluted through PMMA bone cement [14]. An ideal drug delivery system should provide adequate antifungal concentrations at the target site, offer a slow and sustained release of an antimicrobial over an extended period, and be biodegradable so that a second operation is not needed.

Biodegradable antifungal-agent-loaded beads possess advantages over conventional PMMA beads in four ways. First, biodegradable beads provide high concentrations of antifungal agents for the extended time needed to completely treat the particular orthopedic infection. Second, variable biodegradability from weeks to months permits various types of infections to be treated. Third, the biodegradable vehicles degrade eventually, and surgical removal of the beads is not required. Fourth, the biodegradable beads dissolve gradually and the soft tissue or bone defect slowly fills with tissue, it is thus not necessary for bone/tissue reconstruction [15].

This current study developed biodegradable antifungal-agent-loaded vehicles for a long-term drug release. We utilized a compression-molding method to fabricate fluconazole-incorporated Poly(d,l-lactide-*co*-glycolide) (PLGA) beads and assessed drug release dynamics in vitro and in vivo. Among the various polymeric materials available to develop local drug release systems, PLGA is promising, owing to its degradable polymer that facilitates long-term drug delivery at high doses to the target region [16,17]. Moreover, this material has been certified for clinical use owing to its nontoxic nature and minimum inflammatory effects.

After molding, bead morphology was evaluated via scanning electron microscopy (SEM), and intrabead antifungal drug incorporation was assessed via Fourier-transform infrared (FTIR) spectroscopy. In vitro fluconazole liberation curves of PLGA beads were inspected via high-performance liquid chromatography (HPLC), and in vivo fluconazole release was investigated by implanting the drug-incorporated beads into the bone cavity of rabbits. We found that a level of fluconazole greater than the minimum therapeutic concentration (MTC) was released for more than 49 d in vivo, indicating that compression-molded PLGA/fluconazole beads are a promising candidate for the treatment of bone infections.

## 2. Materials and Methods

### 2.1. Fabrication of Poly(d,l-lactide-co-glycolide) PLGA/Amphotericin B and PLGA/Fluconazole Beads

All materials utilized in this study, including PLGA (50:50), amphotericin B, and fluconazole, were acquired from Sigma-Aldrich (St. Louis, MO, USA).

PLGA/amphotericin B and PLGA/fluconazole beads were prepared using a laboratory-scale compression-molding system equipped with an isothermal oven with a temperature range of 25 °C to 300 °C. Beads of two different polymer:drug ratios (6:1 and 4:1) and three sizes (3, 5, and 8 mm) were fabricated. PLGA, amphotericin B, and fluconazole at predetermined weights were first mixed using a dry mixer and placed in molds (Figure 1) customized for this study. Table 1 lists the composition of beads of different sizes. The mold, along with the mixture, was then compressed at 700 MPa and placed in an isothermal oven at 65 °C for sintering for 1.5 h. Pure PLGA beads were simultaneously prepared as a control.

### 2.2. In Vitro Analysis of Amphotericin B and Fluconazole Release

In vitro release of amphotericin B and fluconazole from the biodegradable beads was assessed via an elution method. PLGA/amphotericin B and PLGA/fluconazole beads were placed in test tubes (*N* = 3) containing 1 mL buffered solution at 37 °C. The tubes were deposed in an isothermal oven for 24 h, and the eluent was harvested and substituted with fresh solution (1 mL). The process was carried out in duplicate for 40 d. Amphotericin B and fluconazole levels in the harvested eluents were quantified via HPLC, carried out using a Hitachi L-2200 System (Hitachi Medical Systems, Tokyo, Japan). All experiments were performed in triplicate (*N* = 3).

### 2.3. FTIR Spectrometry

The thermal stability of fluconazole was determined via FTIR spectrometry to evaluate whether the chemistry and orientation of material structures varied with temperature. FTIR analysis was conducted using a Nicolet iS5 spectrometer (Thermo Fisher Scientific, Waltham, MA, USA) at a resolution of 4 cm^−1^ (32 scans). The drugs were compressed as KBr discs, and spectra were recorded over 400–4000 cm^−1^. Fluconazole was considered stable if no obvious structural changes occurred due to temperature. The FTIR spectra of pure PLGA beads were compared with those of fluconazole-loaded PLGA beads.

### 2.4. Determination of the Water Contact Angle

The water contact angles of the beads were measured on a contact angle measurement device (First Ten Angstroms, Portsmouth, VA, USA) (*N* = 3).

### 2.5. Cell Culture

Cytotoxicity of fluconazole-loaded PLGA beads was examined via a Cell Counting Kit-8 (CCK-8) assay (Sigma-Aldrich, St. Louis, MO, USA) for cell viability, in accordance with the manufacturer’s instructions. Eluents harvested at 1, 2, 3, 7, and 14 d were placed in 96-well culture plates. Human fibroblasts obtained from foreskins of patients (1–3 years of age) undergoing surgery were seeded (1 × 10^4^ cells/well) in Dulbecco’s Modified Eagle’s Medium (DMEM) at 37 °C and 5% CO_2_/95% for 48 h. Cell viability was monitored via the CCK-8 assay and quantified using an ELISA reader.

### 2.6. In Vivo Animal Study

All animal experimental procedures were approved by the Institutional Animal Care and Use Committee of Chang Gung University (CGU107-275, Approved 19 March 2019), and all experimental animals were provided care in accordance with the regulations of the Ministry of Health and Welfare of Taiwan under the supervision of a licensed veterinarian.

Four adult New Zealand white rabbits (Animal Health Research Institute, Panchiao, Taiwan) weighing approximately 3.5 ± 0.3 kg were enrolled in the experiment. The rabbits were housed in individual cages in a temperature- and light-controlled room, with ad libitum access to standard rabbit chow and sterilized drinking water. All animals were administered general anesthesia via inhalation of isoflurane through a vaporizer (Matrx, Pompano Beach, FL, USA) in a plastic box (40 cm × 20 cm × 28 cm). Anesthesia was maintained during the entire surgical procedure via mask inhalation of isoflurane.

After rabbits were sedated, the right femoral sites were depilated, washed with soft soap, and treated aseptically with 70% ethanol directly before the surgical procedure. The other site of the animals was covered with a sterile blanket. Under aseptic conditions, the middle/third region of the right femur was dissected via an anterolateral approach. A bone defect (5.0 × 10.0 mm^2^) was induced at the right femoral middle site, and a polymethylmethacrylate (PMMA) spacer was initially inserted. The wound was closed with 3-0 Vicryl sutures (Johnson & Johnson, New Brunswick, NJ, USA). After 2 weeks, the PMMA spacer was surgically excised and fluconazole-impregnated PLGA cylindrical beads (5.0 × 6.0 mm^2^) were placed into the right femoral bone cavity (Figure 2). The wound was closed in a layer-by-layer manner. In vivo drug concentrations were determined by sampling specimens (surgical site fluid) with aspirates obtained on days 1, 2, 3, 7, and 14. After 2 weeks, owing to difficulty aspirating fluid from the surgical site, we harvested tissue around the bead area on days 21, 28, 35, 42, and 49. Cylindrical specimens of tissue surrounding the beads were centrifuged, and the plasma was sampled and stored at −80 °C until analysis. In vivo fluconazole concentrations in the tissue samples were determined via HPLC. All samples were diluted with phosphate-buffered saline and assessed in accordance with the assay standard curve. A calibration curve was generated for each set of measurements (correlation coefficient > 0.99). Blood samples were obtained from the marginal ear vein using a syringe to determine serum aspartate transaminase (AST) levels and antibiotic concentrations after implanting the fluconazole-impregnated beads. AST levels were determined using the IDEXX Catalyst DX system (Westbrook, ME, USA), while fluconazole concentrations were determined via HPLC. Data thus obtained were used to assess the liver function of rabbits during experiments.

## 3. Results

### 3.1. Characterization of Fabricated Poly(d,l-lactide-co-glycolide) PLGA/Amphotericin B and PLGA/Fluconazole Beads

Amphotericin B- and fluconazole-containing beads of 3, 5, and 8 mm were prepared through compression molding (Figure 3). To confirm that the molding temperature did not deactivate the drugs, a thermal stability test for fluconazole was carried out using FTIR spectroscopy. As shown in Figure 4, amphotericin B and fluconazole remained intact at 70 °C, indicating that a temperature of 65 °C is optimal for compression molding.

Water contact angles determined herein are shown in Figure 5. While the pure PLGA beads exhibited hydrophobic properties (water contact angle of 97.38°), antifungal drug-containing beads were hydrophilic (all angles were less than 70°). In addition, the water contact angle of PLGA/fluconazole beads decreased with an increase in the drug content of beads, primarily owing to the hydrophilic nature of fluconazole.

To confirm successful incorporation of amphotericin B and fluconazole in the beads, FTIR spectra of drug-loaded beads were compared with those of pure PLGA beads (Figure 6). The new absorption peak at 1600 cm^−1^ might be attributable to the C=N bonds of amphotericin B and fluconazole [18,19]. Enhanced absorption at 1670–1780 cm^−1^ corresponded to C=O bonds, primarily owing to supplementation with antifungal drugs, and an absorbance peak of approximately 1270 cm^−1^ may have resulted from C–F bond enhancement in loaded drugs. The FTIR spectra indicate that the antifungal drugs were successfully incorporated into PLGA beads.

### 3.2. In Vitro Release Dynamics of Poly(d,l-lactide-co-glycolide) PLGA/Amphotericin B and PLGA/Fluconazole Beads

Figure 7 illustrates the release dynamics of amphotericin B and fluconazole from antifungal-drug-loaded beads with different polymer:drug ratios (6:1 and 4:1). Drug release was slightly less than 1% of the total release of amphotericin B. Triphasic liberation curves were generated, displaying blast release on the first day, followed by gradual elution on days 2–23, and accelerated drug release on days 23 and 10 in beads measuring 3, 5, and 8 mm, respectively. Beads with a higher drug loading ratio (i.e., polymer:drug = 4:1) generally exhibited greater antifungal drug concentrations in the eluent. Figure 8 shows the release dynamics of PLGA/fluconazole beads of different sizes. Larger beads exhibited an earlier accelerated release of fluconazole than smaller beads. All antifungal-drug-embedded beads exhibited sustained release of fluconazole for more than 30 d.

Cytotoxicity analysis was performed using CCK-8 assays to measure cell viability. The eluent from 5 mm beads with a 4:1 polymer:drug ratio was analyzed. Figure 9 shows that cell viability was reduced on day 1, probably owing to burst release of fluconazole, thereby potentially affecting cell proliferation. Thereafter, PLGA/fluconazole beads showed no signs of cytotoxicity.

### 3.3. In Vivo Drug Release

Owing to the poor release dynamics of amphotericin B, only PLGA/fluconazole beads were selected for in vivo analysis of drug release. Figure 10 shows drug concentrations as measured in bone cavity tissue and blood. Fluconazole-embedded beads displayed long-term fluconazole release (beyond the MTC) for more than 49 d in vivo. Furthermore, blood drug concentration was significantly lower than that in bone tissue.

Blood AST levels were within the physiological range (Figure 11) of 33–99 (U/L) [20].

## 4. Discussion

Treatment of fungal osteomyelitis is more complicated than that of chronic osteomyelitis. Local antibiotic release is important in treating chronic osteomyelitis; however, the use of antifungal-loaded bone cement beads to treat fungal osteomyelitis remains controversial, mainly owing to the inconsistent release dynamics of antifungal drugs from the PMMA beads. While some studies have reported the successful eradication of fungal osteomyelitis or periprosthetic joint infections [21], others have yielded conflicting results. During production of PMMA beads containing amphotericin B or fluconazole, covalent crosslinkage may result in poor drug release from the bone cement [13,14]. The potential risk of irritation of host tissues and the non-biodegradable nature of PMMA potentially limit its applications in treating osteomyelitis, and therefore, surgical excision is often performed as an alternative. This study reports the successful production of antifungal-agent-loaded PLGA beads with amphotericin B or fluconazole using the compression-molding method. In vitro analysis revealed that the elution rate of amphotericin-B-loaded PLGA beads with different ratios did not exceed 1% for 28 d (Figure 7). Moreover, amphotericin-B-loaded PLGA beads displayed poor sustained drug release. However, the fluconazole-loaded PLGA beads with different ratios displayed an elution rate above 90% for 28 d. Based on these results, we selected PLGA/fluconazole beads for in vivo analysis.

Few studies have evaluated the efficacy of antifungal-impregnated PLGA carriers [22,23,24], although the release profile of amphotericin B with an organic solvent was shown to be promising. In the present study, PLGA/amphotericin B beads showed poor release dynamics [25]. The temperature during compression molding was increased up to 65 °C, which is also the temperature used for polymerization of amphotericin-impregnated PMMA beads, before sintering for 1.5 h. The polymerization process may have facilitated covalent linkages in amphotericin-impregnated PMMA beads; however, the actual mechanism resulting in the poor release profile of amphotericin-impregnated PLGA beads remains unclear. Nonetheless, poor release was not observed with fluconazole-impregnated PLGA beads.

Biodegradable PLGA/fluconazole beads released antifungal drugs at high concentrations for over 49 d, thus controlling the bone infection. The present work is the first study to develop biodegradable antifungal beads using a compression-molding technique without the use of organic solvents and to evaluate the sustained release of high and local fluconazole concentrations in vivo. Owing to the absence of organic solvents during bead preparation, these fluconazole/PLGA beads are potentially applicable for clinical use for the treatment of fungal infections in bone tissue. The present results show that most drugs were absorbed by the surrounding tissues, while the systemic drug concentration remained low (Figure 10).

Fluconazole is a first-generation triazole antifungal agent used to treat fungal infections through both oral and parental routes. Fluconazole is metabolized in the kidneys, unlike other azoles metabolized in the liver. Adverse effects have been commonly reported with use of long-term fluconazole therapy [26], and although none of these have been severe or life-threatening [19], liver damage has been observed in some cases [27,28]. In a previous study, elevation of liver transaminases was more common than liver damage, and the incidence of treatment termination owing to elevated liver enzymes was 0.7% [28]. In the present study, hepatoxicity potentially resulting from fluconazole in the blood during the treatment period was assessed upon local administration at high doses. Blood samples were collected from the marginal ear vein to quantify blood AST levels after each biopsy. Louie et al. reported an optimal fluconazole therapeutic concentration of not more than 10 mg/kg [29]. In the present study, fluconazole was used at 10 mg/kg (a cylindrical bead containing 35 mg fluconazole/3.5 kg body weight), and blood AST levels were within the physiological range in all the New Zealand white rabbits used in the study. These results indicate that fluconazole-impregnated PLGA beads are safe (Figure 11).

PLGA is one of the most suitable biodegradable polymeric materials for synthesizing drug delivery devices and for tissue engineering [30,31,32]. The material is biocompatible and biodegradable, exhibits a wide range of erosion times, and is mechanically tunable. Most importantly, PLGA is an FDA-approved polymer used extensively for controlled release of small-molecule drugs, proteins, and other macromolecules. Therefore, we selected PLGA as the carrier material for antifungal fluconazole-containing beads.

Drug release from a biodegradable carrier generally occurs in three stages: primary blast, diffusion-dominated elution, and degradation-dominated release [15]. After compression molding, most drugs are dispersed into the volume of the PLGA/fluconazole beads; however, certain drug formulations on the particle surface may lead to an initial drug release burst, followed by controlled drug release by diffusion and other factors. Relatively constant slow elution of the antifungal agents was thus observed. Finally, PLGA/fluconazole beads swell owing to water uptake during elution, thus damaging the polymer matrix and forming openings for antifungal release. The rate of fluconazole release thus accelerated accordingly.

Sustained local release of high levels of antifungals contributes to infection control. Louie et al. reported that the minimum inhibitory concentration median (MIC_50_) in a designated fluconazole mid-resistant infection was 64–128 μg/mL. In a fluconazole-susceptible strain of *Candida*, the median MIC was 0.5 μg/mL [29], and the concentration of released fluconazole was greater than 128 μg/mL. These results demonstrate that biodegradable antifungal-embedded beads can release high concentrations of fluconazole (well beyond the MIC_50_) for more than 49 d (Figure 10). Furthermore, FTIR analysis suggests that bead-embedded drug formulations remain stable during the molding process.

Although the current study has generated promising preliminary data, some limitations should be noted. First, we used a non-infected animal model, and therefore it is unclear whether the antifungal beads might perform differently in infected tissue. Further evaluation of the antifungal agent-embedded PLGA copolymer beads in an animal model of fungal infection is necessary to address this limitation. Second, despite the experimental data showing that 80% of the drug was released at the end of the study, it would have been faster given larger amounts of liquid. The sink condition analysis is needed to ensure the drug was released freely. Third, although no obvious sign of inflammation was observed in the in vivo test, the influence of controlled release of drugs and carriers on the local irritation should be further examined. Finally, the relevance of the present findings to patients with bone infections remains unclear and warrants further investigation. We intend to further explore these topics in future studies.

## 5. Conclusions

This study is the first to perform in vitro and in vivo analyses of drug release dynamics in the bone cavity from compression-molded antifungal-incorporated PLGA beads prepared without organic solvents. Biodegradable PLGA/fluconazole beads released high concentrations of antifungals for over 49 d at the target site, while the antifungal agent blood concentration remained low. In summary, the compression-molded PLGA/fluconazole beads that we describe here have potential applications for treating bone infections.

## Figures and Tables

**Figure 1 pharmaceutics-11-00550-f001:**
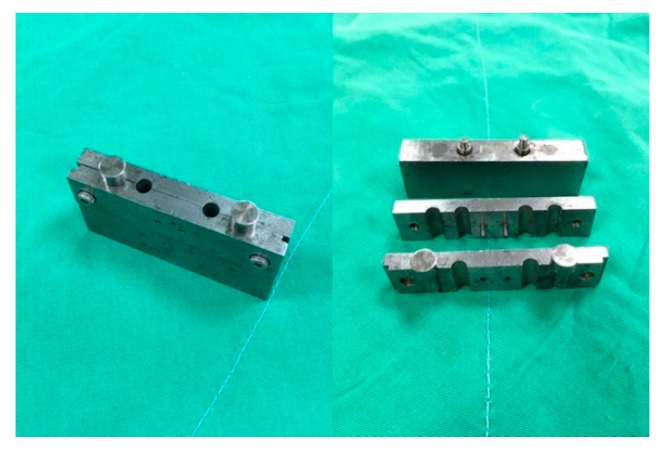
Mold used for compression molding of the antifungal-agent-containing beads.

**Figure 2 pharmaceutics-11-00550-f002:**
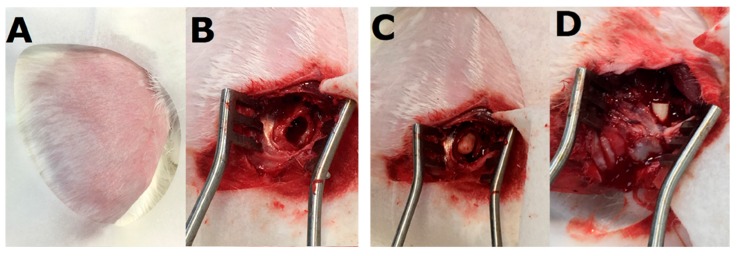
Images of the surgical procedure. (**A**) The right femoral site was depilated and sterilized. (**B**) a bone cavity was formed (5.0 × 10.0 mm^2^); (**C**) a polymethylmethacrylate spacer was placed into the bone cavity; (**D**) after 2 weeks, fluconazole-impregnated Poly(d,l-lactide-*co*-glycolide) cylindrical beads (5.0 × 6.0 mm^2^) were placed into the right femoral bone cavity.

**Figure 3 pharmaceutics-11-00550-f003:**
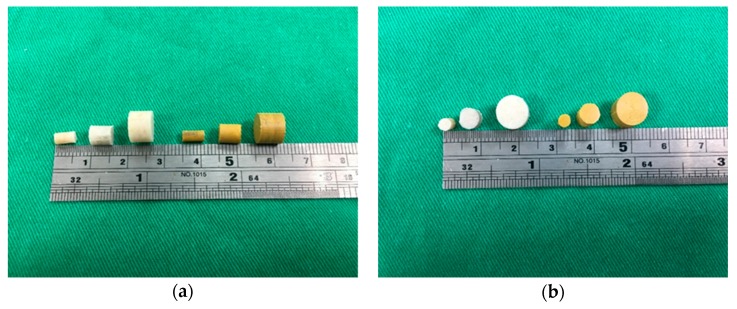
Photographs of the fabricated drug-loaded beads. The white cylindrical beads are fluconazole-impregnated Poly(d,l-lactide-*co*-glycolide) (PLGA) beads. The yellow cylindrical beads are amphotericin B-impregnated PLGA beads.

**Figure 4 pharmaceutics-11-00550-f004:**
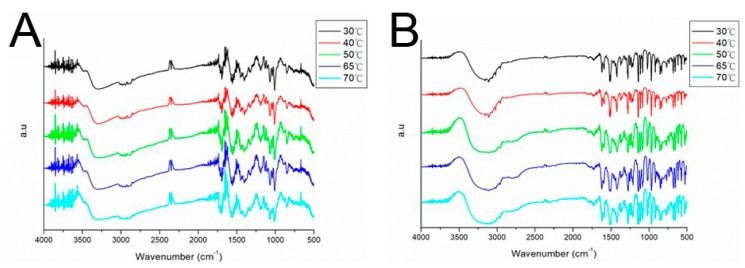
Effect of molding temperature on (**A**) amphotericin B and (**B**) fluconazole stability determined by Fourier-transform infrared spectroscopy.

**Figure 5 pharmaceutics-11-00550-f005:**
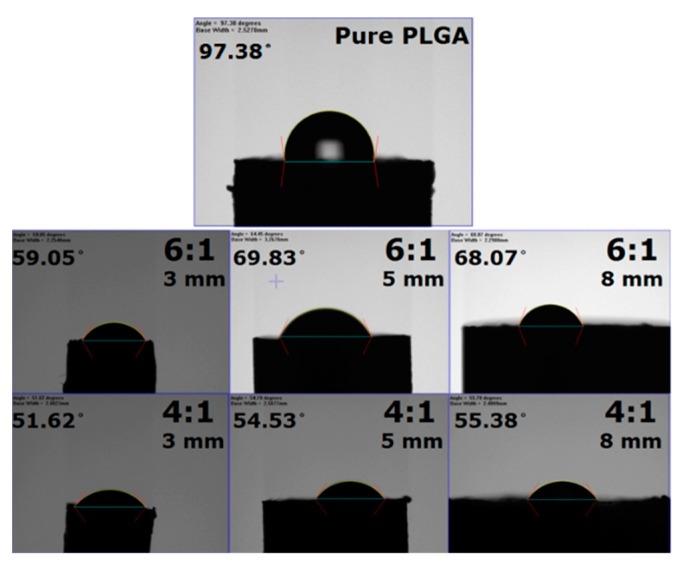
Water contact angles for pure Poly(d,l-lactide-*co*-glycolide) and drug-loaded beads of different polymer:drug ratios and sizes.

**Figure 6 pharmaceutics-11-00550-f006:**
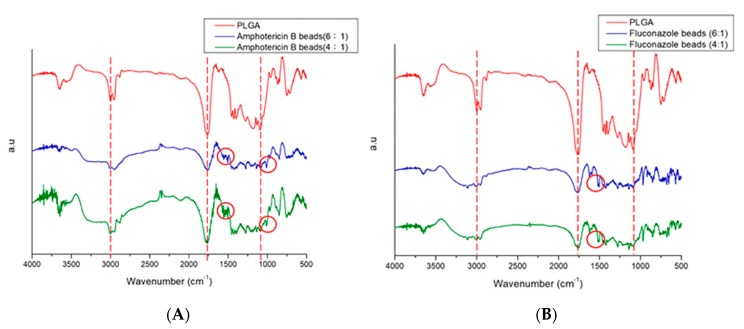
Fourier-transform infrared spectra of (**A**) pure Poly(d,l-lactide-*co*-glycolide) (PLGA) and amphotericin B/PLGA beads (**B**) pure Poly(d,l-lactide-*co*-glycolide) (PLGA) and fluconazole/PLGA beads.

**Figure 7 pharmaceutics-11-00550-f007:**
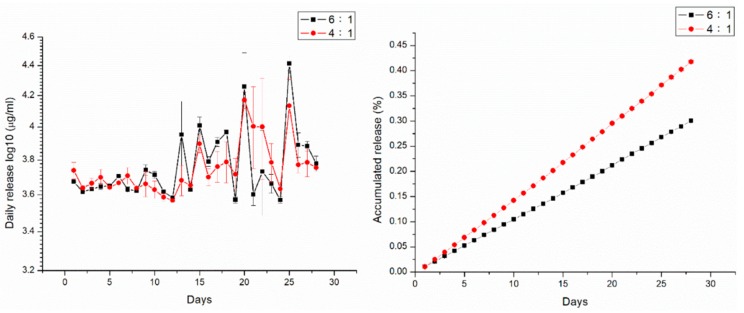
In vitro release curves of amphotericin B from the drug-loaded beads.

**Figure 8 pharmaceutics-11-00550-f008:**
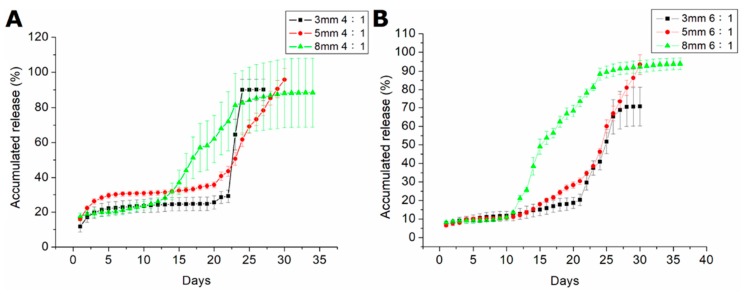
In vitro release curves of fluconazole from drug-loaded beads with various polymer:drug ratios. (**A**). The accumulated release of beads of different sizes with 4:1 polymer:drug ratios. (**B**). The accumulated release of beads of different sizes with 6:1 polymer:drug ratios.

**Figure 9 pharmaceutics-11-00550-f009:**
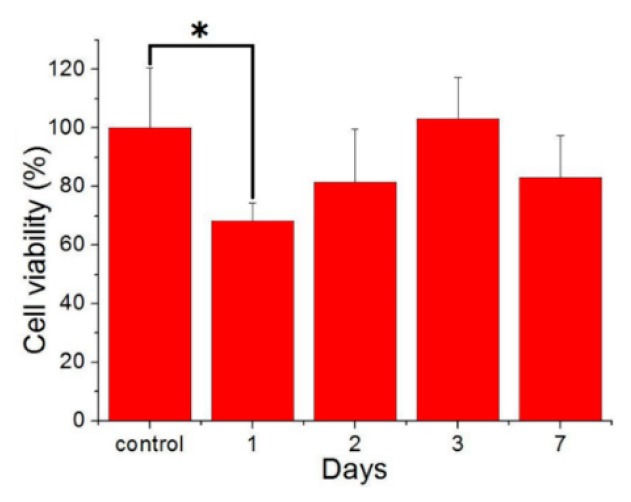
Cell viability of the fluconazole-incorporated beads (* *p* < 0.05).

**Figure 10 pharmaceutics-11-00550-f010:**
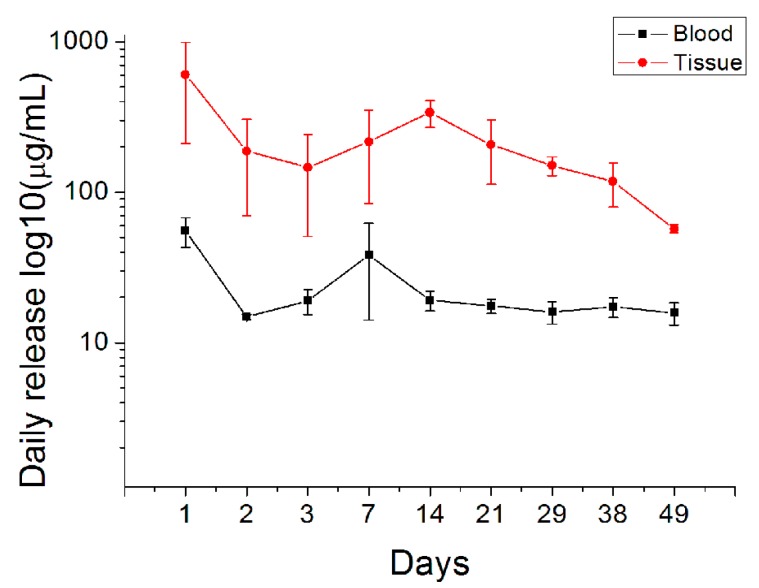
In vivo release curves of fluconazole from the drug-loaded beads.

**Figure 11 pharmaceutics-11-00550-f011:**
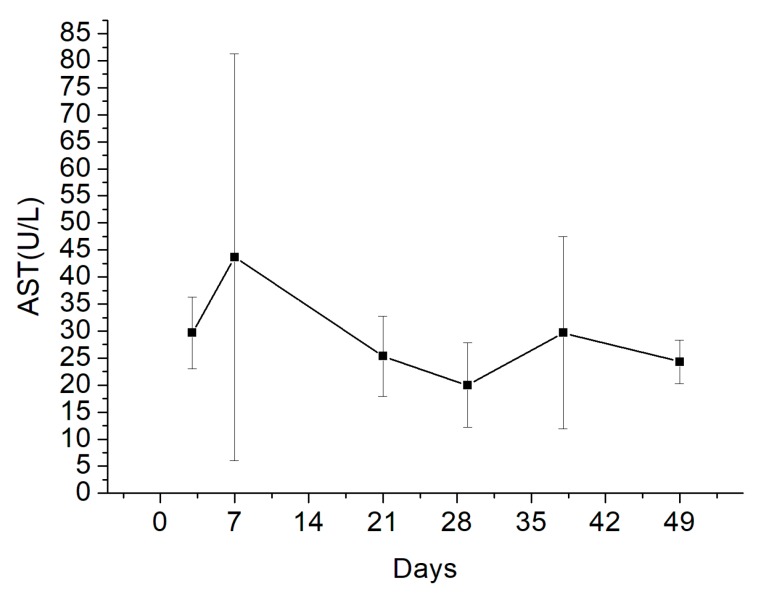
Estimation of blood aspartate transaminase (AST) levels.

**Table 1 pharmaceutics-11-00550-t001:** Composition of the fluconazole-loaded Poly(d,l-lactide-*co*-glycolide) beads.

PLGA/Drug Ratio	Size: Diameter × Height (mm × mm)	Weight of PLGA (mg)	Weight of Drug (mg)
6:1	3 × 5.65	54	9
	5 × 5.55	150	25
	8 × 6.18	384	64
4:1	3 × 5.02	50.4	12.6
	5 × 6.27	140	35
	8 × 6.25	358.4	89.6

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
