# Peer review of "Resorbable Beads Provide Extended Release of Antifungal Medication: In Vitro and In Vivo Analyses"

_pharmaceutics, 2019, doi:10.3390/pharmaceutics11110550_

Round 1

Reviewer 1 Report

In this manuscript, Hsu et.al designed the antifungal drug-loaded PLGA beads as a potential platform to treat fungal osteomyelitis. Beads with different size and drug content were evaluated in vitro and in vivo. The drug encapsulation and stability after molding was confirmed. The beads were able to provide controlled and sustained drug release over 30 days at both in vitro and in vivo condition without causing noticeable toxicity. Given this is a complete and substantial work, there are some major issues with the experiment design, data presentation, which needs a major revision before it can be considered for publication.

In the introduction, the author gave a good review of the current treatment method and related challenges that impaired the efficacy of antifungal-agent-impregnated beads. However, the goal for this work is not clearly presented. (Line 59) And they didn’t differentiate their platform from other available platforms, such as PMMA and PLGA microparticles. A major issue with the drug release study in vitro is the lack of sink condition. Both amphotericin B and fluconazole are considered as drug that are not very soluble in water. However, the authors did the drug release in 1 mL of buffer solution, the buffer will be easily saturated by the released drug and the further release will be hampered. Without repeating the drug release in a larger volume of the buffer, the release data of both drug-impregnated beads are not valid. Is the blood AST level a valid parameter to evaluate the toxicity? I assume the authors are more interested in the local irritation generated by the drug loaded implant. Then it would more relevant to examine the histological signature and cytokine secretion from local tissue. Other minor points: Please delete: “For research articles with several authors, a short paragraph specifying their individual contributions must be provided. The following statements should be used” at line 301 Please delete: "Please add: “This research received no external funding”" at line 310.

Author Response

Reviewer #1 (Comments for the Author):

In the introduction, the author gave a good review of the current treatment method and related challenges that impaired the efficacy of antifungal-agent-impregnated beads. However, the goal for this work is not clearly presented. (Line 59)

Authors’ response

We have revised the manuscript to better present the purpose of this study. “The study aimed to develop biodegradable antifungal agents loaded vehicles for the treatment of chronic fungal osteomyelitis. “(Revision edition: Lines 60-61)

 And they didn’t differentiate their platform from other available platforms, such as PMMA and PLGA microparticles.

Authors’ response

The manuscript has been revised to discuss the difference of our PLGA beads from antifungal PMMA carriers in the discussion section. “Local antibiotic release is important in treating chronic osteomyelitis; however, the use of antifungal-loaded bone cement beads to treat fungal osteomyelitis remains controversial, mainly owing to the inconsistent release dynamics of antifungal drugs from the PMMA beads.” (Revision edition: Lines 241-244)

We have also discussed the difference of our PLGA beads from other PLGA microspheres in the discussion section. “The present work is the first study to develop biodegradable antifungal beads using compression-molding technique, without the use of organic solvents, and to evaluate the sustained release of high and local fluconazole concentrations in vivo. Owing to the absence of organic solvents during bead preparation, these fluconazole/PLGA beads are potentially applicable for clinical use for the therapy of fungal infections in bone tissue.” (Revision edition: Lines 267-271)

A major issue with the drug release study in vitro is the lack of sink condition.

Authors’ response:

We did not carry out the sink condition about the fluconazole loaded PLGA beads. Nevertheless, the experimental results of fluconazole/PLGA beads demonstrated that nearly 90% of total fluconazole amount is released from the beads in this study. This suggests that the daily replacement of PBS can effectively represent the release characteristics.

Both amphotericin B and fluconazole are considered as drug that are not very soluble in water. However, the authors did the drug release in 1 mL of buffer solution, the buffer will be easily saturated by the released drug and the further release will be hampered. Without repeating the drug release in a larger volume of the buffer, the release data of both drug-impregnated beads are not valid.

Authors’ reponse:

Previous studies have employed the same in vitro elution method to determine the release behaviors of amphotericin B and fluconazole. In Khan et al.’s study, one mL of sterile 20 mM PBS was added to each vial and the aliquots were then incubated at 25°C. Release runs were monitored for 8 days. 100 μL aliquot was taken out from each sample daily and centrifuged at 10,000g for 10 min. (A.A. Khan et al. European Journal of Pharmaceutical Sciences 119 (2018) 171–178) In Sealy et al.’s research, they used the PBS to replace the solution of bioactive and nonbioactive bone cement with amphotericin B and ceftazidime to investigate the release profiles. (PI Sealy et al. Ann Pharmacother 2009;43:1606·15) Souza et al. investigated the release profile of amphotericin B from nanoparticles and also employed the PBS to replace the solution. (A.C.O. Souza et al. European Journal of Medicinal Chemistry 95 (2015) 267e276) This study therefore used the same scheme to study the release characteristics of antifungal drugs.

Is the blood AST level a valid parameter to evaluate the toxicity? I assume the authors are more interested in the local irritation generated by the drug loaded implant. Then it would more relevant to examine the histological signature and cytokine secretion from local tissue.

Authors’ response:

Liver damage has been reported in some cases with the use of fluconazole. (Tverdek, F.P., Expert Rev. Anti. Infect. Ther. 2016, 14, 765–776, Wang, J.L. et al., Antimicrob. Agents Chemother. 2010, 54, 2409–2419) In other study, the elevation of liver transaminases was more commonly seen than liver damage and the incidence of treatment termination owing to elevated liver enzymes was 0.7%. (Wang, J.L. et al., Antimicrob. Agents Chemother. 2010, 54, 2409–2419) In this work, the hepatoxicity potentially resulting from fluconazole in the blood during the treatment period was assessed upon local administration at high doses. The experimental results demonstrated high concentrations of fluconazole not only in the local area but also in the blood sample. Therefore, we measured the blood AST level to verify the safety in adopting fluconazole of sustained release and high drug concentration.

Other minor points:

Please delete: “For research articles with several authors, a short paragraph specifying their individual contributions must be provided. The following statements should be used” at line 301 Please delete: "Please add: “This research received no external funding”" at line 310.

Authors’ response:

Thanks for your kind suggestion and we have deleted the sentences: “For research articles with several authors, a short paragraph specifying their individual contributions must be provided. The following statements should be used” (Revision edition: Line 324) and "Please add: “This research received no external funding”" (Revision edition: Line 332).

Reviewer 2 Report

The manuscript reports a study on amphotericin- or fluconazole-incorporated poly(d,l-lactide-co-glycolide)(PLGA) beads. Beads of two different polymer:drug ratios (6:1 and 4:1) and three sizes (3, 5, and 8 mm) were fabricated by compression molding a 65°C without the use of organic solvents. The paper is well organized and the results are clearly presented.

The successful incorporation of fluconazole in the beads was confirmed by FTIR spectra, suggesting also that the bead-embedded drug formulation remained stable during the molding process. Why the authors did not report the same analysis also for amphotericin-loaded PLGA beads?

Furthermore, the abstract lacks of amphotericin… please, complete the sentence: “This study aimed to evaluate in vitro and in vivo liberation patterns of fluconazole-incorporated poly(d,l-lactide-co-glycolide) (PLGA) beads.”

Release studies were performed in vitro and in vivo, implanting the fluconazole-incorporated beads into the bone cavity of rabbits.

In my opinion, the paper is acceptable for publication.

Author Response

Reviewer #2 (Comments for the Author):

The manuscript reports a study on amphotericin- or fluconazole-incorporated poly(d,l-lactide-co-glycolide)(PLGA) beads. Beads of two different polymer: drug ratios (6:1 and 4:1) and three sizes (3, 5, and 8 mm) were fabricated by compression molding a 65°C without the use of organic solvents. The paper is well organized and the results are clearly presented.

Authors’ response:

Thank you for the positive comments!

The successful incorporation of fluconazole in the beads was confirmed by FTIR spectra, suggesting also that the bead-embedded drug formulation remained stable during the molding process. Why the authors did not report the same analysis also for amphotericin-loaded PLGA beads?

Authors’ response:

We had conducted the FTIR spectra of amphotericin B as well. However, the poor release profile of amphotericin B was noted later, so we only included the FTIR spectra of fluconazole. The FTIR spectra of amphotericin B has been added and addressed in the revised manuscript (Figure 6A). The manuscript was revised with “To confirm successful incorporation of amphotericin B and fluconazole in the beads, FTIR spectra of drug-loaded beads were compared with those of pure PLGA beads (Figure 6)” (Revision edition: Lines 186-188) and the legend description was also revised. “Figure 6: Fourier-transform infrared spectra of (A) pure poly(d,l-lactide-co-glycolide) (PLGA) and Amphotericin B/PLGA beads (B) pure poly(d,l-lactide-co-glycolide) (PLGA) and fluconazole/PLGA beads” (Revision edition: Lines 195-197)

Furthermore, the abstract lacks of amphotericin… please, complete the sentence: “This study aimed to evaluate in vitro and in vivo liberation patterns of fluconazole-incorporated poly(d,l-lactide-co-glycolide) (PLGA) beads.”

Authors’ response:

Thank you for the comments. The sentence has been revised to “This study aimed to evaluate the liberation patterns of amphotericin B- and fluconazole-incorporated poly(d,l-lactide-co-glycolide) (PLGA) beads.” (Revision edition: Lines 15-17).

Reviewer 3 Report

The innovation of this paper is a big problem. Using PLGA loading two insoluble drugs for bulk material control release is not a good idea. The paper lack the evidence that whether the beads mechanical property would influence the drug diffuse. This paper also lacking the most important pharmacokinetics analysis part based on drug release. The language is another weakness of this paper. Based on these points, I am not suggesting this paper to be published on Pharmaceutics.

Author Response

Reviewer #3 (Comments for the Author):

The innovation of this paper is a big problem. Using PLGA loading two insoluble drugs for bulk material control release is not a good idea. The paper lack the evidence that whether the beads mechanical property would influence the drug diffuse. This paper also lacking the most important pharmacokinetics analysis part based on drug release.

Authors’ response:

Thank you for the comments. Fungal osteomyelitis has been difficult to treat in current clinical condition. Prolong antifungal therapy with good compliance is important to the successful treatment. (Ueng et al., Clin Orthop Relat Res (2013) 471:3002–3009) In this study, we did not load two drugs into the beads at the same time. The amphotericin B-loaded beads and fluconazole-loaded beads were prepared and analyzed separately.

Increasing the compression pressure generally increases the mechanical property and density of compression molded beads (Liu et al., J Biomed Mat Res (1999) 48:613-620), and the total period of drug release can be improved. We investigated the in vitro and in vivo releases of pharmaceuticals from the biodegradable beads, and found that a high level of fluconazole (beyond the minimum therapeutic concentration [MTC]) was released for more than 49 days in vivo. Our results demonstrate that compression-molded PLGA/fluconazole beads have potential applications in treating bone infections.

The language is another weakness of this paper.

Authors’ response:

Thank you for the comments. The manuscript has been proofed by professional editing-service to improve the readability. The certification is shown as follows.

Round 2

Reviewer 1 Report

It's great that the authors add a sentence to explain the goal of the study. However, the motivation is not clear enough. If like the author said, the motivation was the drug release studies with PMMA system are controversial, I would study the reason behind instead of starting with a new drug release platform. The author need to restructure the introduction part with more reasonable motivation.  The sink condition is to ensure the drug was released freely. Although the author found 80% of drug was released at the end of the study, it would have been faster given larger amount of liquid. Thus, without improving the study with higher amount liquid (e.g. 2 liters), the release pattern shown in the manuscript can not reflect the real kinetics .  I agree with the author that AST level could be used to determined the systemic toxicity. Would it be more relevant to test the local irritation? Other minor issues: In line 16, "liberation pattern" is a misused term for drug release.  

Author Response

It's great that the authors add a sentence to explain the goal of the study. However, the motivation is not clear enough. If like the author said, the motivation was the drug release studies with PMMA system are controversial, I would study the reason behind instead of starting with a new drug release platform. The author need to restructure the introduction part with more reasonable motivation. 

Authors’ response:

Thanks for the reviewer’s comment. Kweon et al. reported that adding 10 g poragen to antifungal-loaded bone cement (ALBC) containing 200 mg amphotericin B decreases the compressive strength of PMMA beads and thus limits its use for implant fixation [12]. Sealy et al. showed poor release dynamics of fluconazole in ALBC [13]. Furthermore, Goss et al. reported that amphotericin B could not be eluted through PMMA bone cement [14]. An ideal drug delivery system should provide adequate antifungal concentrations at the target site, offer a slow and sustained release of antimicrobial over an extended period, and be biodegradable so that a second operation is not needed.

Biodegradable antifungal agents loaded beads possess advantages over conventional PMMA beads in four ways. First, biodegradable beads provide high concentrations of antifungal agents for the extended time needed to completely treat the particular orthopedic infection. Second, variable biodegradability from weeks to months permits various types of infections to be treated. Third, the biodegradable vehicles degrade eventually, surgical removal of the beads is not required. Fourth, the biodegradable beads dissolve gradually and the soft tissue or bone defect slowly fills with tissue, it is thus not necessary for bone/tissue reconstruction [15].

This current study developed biodegradable antifungal agents loaded vehicles for a long term drug release.

The manuscript has been revised to better address the motivation of this study (Lines 53-69).

The sink condition is to ensure the drug was released freely. Although the author found 80% of drug was released at the end of the study, it would have been faster given larger amount of liquid. Thus, without improving the study with higher amount liquid (e.g. 2 liters), the release pattern shown in the manuscript cannot reflect the real kinetics. 

Authors’ response:

Thank you for the comments. We recognized the importance of sink condition analysis, and will conduct the work in our future studies. We have revised the manuscript to include this as the limitation of the current study. (Lines 321-323)

I agree with the author that AST level could be used to determine the systemic toxicity. Would it be more relevant to test the local irritation?

Authors’ response:

Thank you for the comments. Indeed, although no obvious sign of inflammation was observed in the in vivo test, the influence of controlled release of drugs and carriers on the local irritation should be further examined. We have revised the manuscript to include this as the limitation of the current study. (Lines 323-325)

Other minor issues: In line 16, "liberation pattern" is a misused term for drug release.  

Authors’ response:

Thanks for your kind suggestion. “liberation” has been revised to “release”. (Line 16)

Reviewer 3 Report

Utilizing the drug control release technique is the trend of the next generation drug delivery to treat disease. The author reported to use PLGA beads as potential tools to treat bone infection. The author have finished many part of this project from the material characterization, drug release, cell evaluation and animal test.

Honestly say, I not prefer this idea is because physically blending to achieve drug control release would cause many negative effects. Though this PLGA beads reported have a two month long-term release profile, how to main drug with a stable therapeutic concentration by this intervention strategy is a very big challenging. I didn’t see author provide any strong pharmaceutical analysis to convince me.

This paper is based on a physical loading method and what interest me is how this PLGA beads shape, density, molding condition would influence drug diffusivity and by improving the release profile and benefit in vivo evaluation. The 700 MPa, 65oC, 1.5hour seems not an optimized condition. I think author lacking more work to make clear of this material. The PLGA degradation would be one influence factor to study the drug release. Author didn’t exhibit any PLGA beads degradation trend influence the drug release.

In vitro evaluation by one cell viability test is too weak. The animal model need to be substitute on a disease model to support this paper.   

Taken together, author have done many research works to exhibit this PLGA beads strategy. Based on my knowledge and recent two issue publications quality on Pharmaceutics, I am so sorry to say that I am not suggesting this paper to be published.

Author Response

Utilizing the drug control release technique is the trend of the next generation drug delivery to treat disease. The author reported to use PLGA beads as potential tools to treat bone infection. The author have finished many part of this project from the material characterization, drug release, cell evaluation and animal test.

Honestly say, I not prefer this idea is because physically blending to achieve drug control release would cause many negative effects. Though this PLGA beads reported have a two month long-term release profile, how to main drug with a stable therapeutic concentration by this intervention strategy is a very big challenging. I didn’t see author provide any strong pharmaceutical analysis to convince me.

This paper is based on a physical loading method and what interest me is how this PLGA beads shape, density, molding condition would influence drug diffusivity and by improving the release profile and benefit in vivo evaluation. The 700 MPa, 65oC, 1.5hour seems not an optimized condition. I think author lacking more work to make clear of this material. The PLGA degradation would be one influence factor to study the drug release. Author didn’t exhibit any PLGA beads degradation trend influence the drug release.

Authors’ response:

Thank you for the comments. Liu et al. [15] fabricated biodegradable vancomycin beads and evaluate the influence of processing conditions including compression temperature, compression pressure, drug to polymer ratio, molecular weights of polymers on the release characteristics of the beads. They found that one can prolong the total effective release period of vancomycin from the beads by adopting a higher molecular weight polymer, increasing the compression pressure, using higher polymer-to-antibiotic ratios, increasing the sintering temperatures, increasing the size of the beads, or making multilayered beads. This literature has been added into the reference list (Ref. [15]) and addressed and cited in the revised manuscript. (Lines 61-67, 303)

In vitro evaluation by one cell viability test is too weak. The animal model need to be substitute on a disease model to support this paper.  

Authors’ response:

Thank you for the comments. The cell viability test has been triplicated to better represent the outcome. In this study, we used a non-infected animal model, and therefore it is unclear whether the antifungal beads might perform differently in infected tissue. Further evaluation of the antifungal agent-embedded PLGA copolymer beads in an animal model of fungal infection is necessary to better answer the question. We have included this in the revised manuscript as the limitation of this study. (Lines 318-321)

Taken together, author have done many research works to exhibit this PLGA beads strategy. Based on my knowledge and recent two issue publications quality on Pharmaceutics, I am so sorry to say that I am not suggesting this paper to be published.

Authors’ response:

Thank you for your comments. The manuscript has been revised to better address the experimental results.

Round 3

Reviewer 1 Report

The authors have addressed my major concerns in the manuscript. The logical flow is much improved too.